# Motivational Orientation in University Athletes: Predictions Based on Emotional Intelligence

**DOI:** 10.3390/bs12100397

**Published:** 2022-10-17

**Authors:** Isabel Mercader-Rubio, Nieves Gutiérrez Ángel, Antonio Granero-Gallegos, Nieves Fátima Oropesa Ruiz, Pilar Sánchez-López

**Affiliations:** 1Department of Psychology, Faculty of Education Sciences, Universidad de Almería, La Cañada, 04120 Almería, Spain; 2Deparment of Education, Faculty of Education Sciences, Universidad de Almería, La Cañada, 04120 Almería, Spain

**Keywords:** university students, motivational orientation, ego, emotional intelligence

## Abstract

While in task orientation, athletes tend to compare themselves to others in order to evaluate their success, in ego orientation athletes have the ability to distinguish between effort and ability and the comparison is made with others in order to evaluate success. The purpose of this study was to inspect the relationship concerning the three dimensions of emotional intelligence (attention, clarity, and emotional regulation) and the two predominant motivational orientations: task-oriented motivation and ego-oriented motivation. The sample was composed of a total of 165 university students from studies within the field of sciences related to sport and physical activity. The main contributions of this research demonstrate the correspondence between emotional intelligence and ego-orientation.

## 1. Introduction

Currently, the analysis of the influence of motivation in sport has been approached by adopting two main theoretical contributions: the achievement goal theory [1] and the self-determination theory [2,3]. In the scientific literature one can find various review papers on achievement goal theory [4] and on self-determination theory [5,6,7].

This research addressed the theoretical contributions of achievement goal theory, from which people act in different contexts motivated by the need to demonstrate their competence. Additionally, it is one of the most widely used theoretical models in the understanding of different variables: cognitive, emotional, and behavioral, all of them related to the achievement of athletes and physical education students [8,9]. According to this theory [10], in achievement contexts, there are two predominant motivational orientations: one task-oriented and the other ego-oriented. Thus, when a student presents a task orientation, they directs their learning objective and conceptualize their own ability level in a circular way, while when it corresponds to the egocentric, the purpose is based on competitiveness and one might conceptualize their skills based on the comparison established with the rest of their environment [1].

Achievement goal theory [11] suggests that people perform different actions based on different expectations and values that they consider necessary to achieve success. These goals may be task-oriented, when they are oriented towards the performance and mastery of a skill, or ego-oriented, when they are oriented towards comparison with others, achieving rewards, avoiding punishment, or complying with external norms [12,13]. It has been shown that the theoretical contribution corresponding to the achievement objectives in the field of sports practice has achieved positive results at a cognitive, behavioral, and emotional level based on student performance [14,15,16].

It is the contributions of the Social Cognitive Theory of Goal Perspectives [1,17] that explain, analyze, and clarify the influence of environmental factors affecting athletes’ achievement motivation. Thus, it is the subject’s own perception that makes the motivational orientation ego-oriented or task-oriented [1,18]. The adoption of one or the other criterion (ego/task) will lead to affective, cognitive, and behavioral consequences, as it is determined by personal characteristics and social and situational aspects [19]. The construct called emotional intelligence refers to the ability of a subject to deal with the emotional communication that they obtain from their environment. Therefore, according to various authors, emotionally intelligent people are those capable of attending to the emotions perceived in their immediate environment, discerning their origin and consequences, and performing optimally in terms of being able to have the ability to regulate the emotional state that provokes them [20,21,22].

Emotional intelligence [23] is defined as “a meta-skill that determines the degree of skill that we can achieve in mastering our other faculties” (p. 68), which means that the subject is capable of recognizing both their own emotions and those of others, manage them, and finally establish relationships [24]. In general terms, emotions involve changes in feelings, cognitions, and behaviors that regulate students’ actions [25,26,27,28]. Therefore, we could highlight that motivation is the necessary impulse to act that occurs as a result of experiencing these emotions, which has been the focus of the scientific community’s interest in recent times [29]. This construct influences students’ cognitive processes and learning strategies as well as their future attitudes towards learning. As a consequence, these high levels of emotional intelligence in athletes [30] help both to lower anxiety and stress levels, throughout the competition process, and to improve sporting and academic performance, and even with life satisfaction [31,32,33,34,35]. Based on the literature reviewed, the aim of this research was to examine the relationship between the three dimensions of emotional intelligence (attention, clarity, and emotional regulation) and the two predominant motivational orientations in university students: task-oriented motivation and ego-oriented motivation.

## 2. Materials and Methods

The methodology corresponds to an investigation that is framed within the so-called retrospective studies, in which the dimensions of emotional intelligence (attention, clarity, and regulation) are confronted with other types of variables, in this case with task-oriented motivation to and ego-oriented motivation. Therefore, it is a retrospective design.

### 2.1. Participants

The sample as a whole was made up of 165 university students who are currently pursuing official undergraduate and postgraduate studies in sports sciences and physical activity. The mean age is 20.33 years, with a standard deviation SD = 3.44. Regarding gender, 70.9% (N = 117) were men and 27.9% (N = 46) women.

### 2.2. Instruments

The first instrument used was the Trait Meta-Mood Scale (TMMS-24) [36] to measure self-perceived emotional capacity. This instrument, through 24 items, assesses each of the dimensions of emotional intelligence on a Likert-type scale (1–5). The psychometric properties of this instrument are adequate in terms of reliability (Cronbach’s alpha), perception (α = 0.90), clarity (α = 0.90), regulation (α = 0.86) and adequate test–retest reliability, perception = 0.60, understanding = 0.70, and regulation = 0.83 [35]. Specifically for this research, Cronbach’s alpha scores = 0.84 were obtained.

The second instrument used is the Perception of Success Questionnaire (POSQ) [37,38], in its Spanish version [39], adapted to physical education [40]. This instrument consists of 12 items and assesses both goal orientation and task orientation on a Likert-type scale (1–10) [41]. The psychometric properties of this instrument are adequate in terms of reliability (Cronbach’s alpha) for items related to task orientation Cronbach’s alpha = 0.80, and for ego orientation Cronbach’s alpha = 0.79. These are similar results to other recent research that has applied the same instrument, which obtained a Cronbach’s alpha = 0.80 for task orientation, and a Cronbach’s alpha = 0.84 for ego orientation [42,43].

### 2.3. Data Analysis

The first data analysis performed corresponds to the descriptive statistics. In this first moment, statistical tests related to the mean, standard deviation, and bivariate correlations were carried out. In a second moment, the reliability index was analyzed and after that, structural equation modeling (SEM) was carried out to test the relationships established in the hypothesized model. Within the structural equation models (SEM), the Jöreskog test was performed in order to determine the covariance structure [44,45] of a multicausal indicator: Multiple Indicators Multiple Causes Model (MIMIC).

To accept or reject this model, the following indices were taken into account [46]: TLI (Tucker–Lewis index), SRMR (standardized mean squared residual), and RMSEA (mean squared error of approximation). Thus, the appropriate indices are: TLI value above 0.95; SRMR values below 0.06; and RMSEA values below 0.08. These analyses were performed with the statistical analysis software SPSS version 26 and R (lavvan, New York, NY, USA).

## 3. Results

Table 1 shows the relationships between each of the dimensions of emotional intelligence: emotional attention, emotional clarity, and emotional regulation (AE/CE/RE) and their relationship with task-oriented motivation (O TAREA) and ego-oriented motivation (O EGO). The following table (Table 1) shows the scores obtained between the study variables, which were positively and reciprocally correlated.

Table 1 shows the relationships between each of the dimensions of emotional intelligence (AE/CE/RE) and their relationship with task-oriented motivation (O TAREA) and ego-oriented motivation (O EGO).

### Structural Equation Model

The hypothetical model of predictive relationships (Figure 1) shows the following indices:-Overall fit indices (evaluate the model overall) and are adequate: *p* < 0.001, RMSEA = 0.00, GFI = 0.99.-Incremental or comparative fit indices (compare the proposed model with the model of independence or absence of relationship between the variables): NNFI= 1.095; TLI = 0.963; CFI = 1; IFI = 1.02.-Parsimony indices (assessing the quality of the model fit in terms of the number of coefficients estimated to achieve that level of fit): AGFI = 0.97.

The results obtained from the structural equation model allow us to establish the following relationships

-There is a positive and direct relationship between emotional intelligence and ego orientation (= 0.048, *p* < 0.001). This allows us to affirm that emotional intelligence is a variable that predicts ego motivation.-There is a relationship that is neither direct nor positive between emotional intelligence and task orientation.-Therefore, based on these results, we can affirm that emotional intelligence is a predictor of ego-motivation.

As can be verified, the results obtained indicate the existence of a direct and positive relationship between emotional intelligence, and each of its three dimensions that it is made up of, with the motivation of the ego: that is, with the fact that a subject performs a sport comparing themselves with the rest, and closely related to obtaining success in the sports field. Therefore, emotional intelligence in these results appears positively and directly correlated with the athlete’s consideration of practicing sports as a social character trait, a key element for obtaining social status.

On the other hand, the results obtained indicate that the same does not occur in the case of emotional intelligence and task motivation. That is, according to the model presented, there would not be a positive and direct relationship between emotional intelligence and that type of motivation that has its ultimate goal in learning and enjoyment itself.

## 4. Discussion

The aim of this research was to analyze the achievement goal theory [1], ego motivation, and task motivation, and their possible links with emotional intelligence and its dimensions: attention, clarity, and emotional regulation. The most substantial findings of this research indicate that there is a relationship between emotional intelligence and ego motivation, understood as that in which the sportsperson performs an action and compares themselves with other people. It is closely linked to the belief that sporting success is achieved by obtaining greater skills than others, and to the consideration of sport as a sign of social status with respect to others, and to a lesser enjoyment of sport [39,40]. That is, from this point of view, success is seen as similar to overcoming rivals [47,48], so many authors consider that a motivational climate related to the ego measures its success through comparison, and not learning [49], being highly concerned with the demonstration of it to others [50].

However, it is also true that other research carried out in the field of sport has highlighted the existence of a direct and positive correspondence between physical self-concept and ego orientation [51]. A plausible explanation for the high levels of ego-orientation in athletes is offered by ref. [52] wherein the authors indicate that factors such as constant evaluation tests, excessive obsession with performance and results, external rewards, or public recognition may explain this fact. On an emotional level, this type of orientation corresponds to high levels of anxiety, hard play, and non-promotion of fair play [53,54].

It should also be noted that different studies that have highlighted the fact that task orientation has high levels in sport contexts that are highly related to competition or professional training [55,56]. In short, both task orientation and ego-orientation allow us to distinguish goals, interests, and emotions related to both success and failure and the incidence of contextual factors in athletes [57,58]. However, we must be cautious with these results obtained due to the number of participants who collaborated with this research. Therefore, we consider that it would be fruitful to reproduce this research with a larger sample size to verify these findings, as this is one of the current limitations of the work.

## 5. Conclusions

The main conclusions reached by this research are as follows:-There is a direct and positive relationship between emotional intelligence and motivation towards the ego.-Such a direct and positive relationship does not occur in the case of the relationship established between emotional intelligence and task orientation-It is valuable to continue research in this field of study, as well as to determine the possible contextual variables that may be influencing such results.

Future investigations will take as thematic lines the analysis of the influence on these results of the sport practiced, or the number of hours dedicated to it. At the same time, they will also intend to analyze whether the degree of improvement or professionalization of the practiced sport influences such results.

## Figures and Tables

**Figure 1 behavsci-12-00397-f001:**
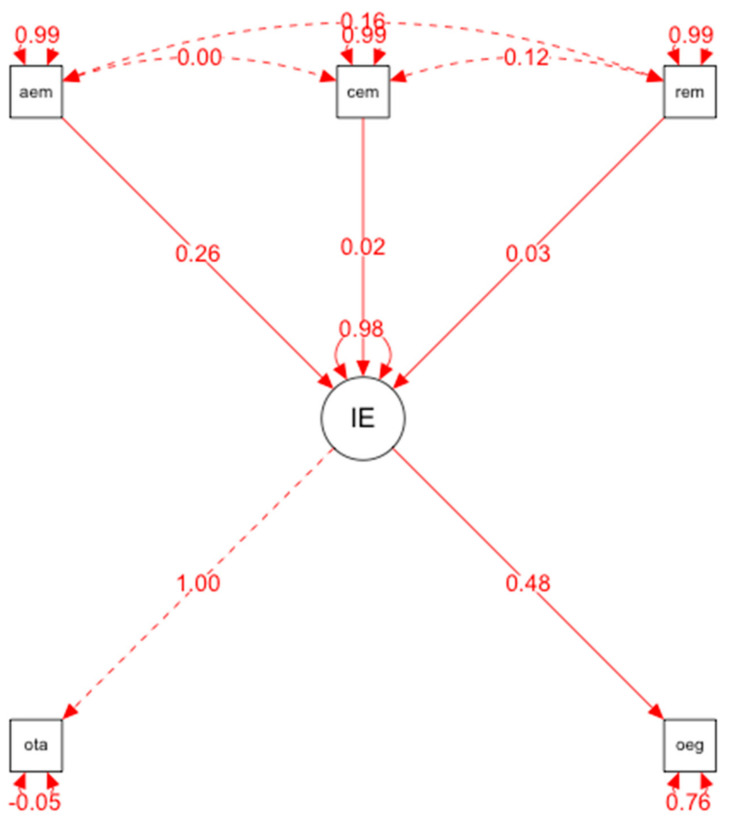
Representation of the results of the structural equation model: MIMIC.

**Table 1 behavsci-12-00397-t001:** Data obtained from the correlations carried out as preliminary analyses.

	1 OTAREA	2 O EGO	3 AE	4 CE	5 RE
1. O TAREA		0.129	0.160 *	−0.043	−0.070
2. O EGO			−0.004	−0.069	0.020
4. AE				0.127	0.263 **
5. CE					0.508 **
6. RE					

Note. * *p* < 0.05; ** *p* < 0.01.

## Data Availability

Not applicable.

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
