# Peer review of "Motivational Orientation in University Athletes: Predictions Based on Emotional Intelligence"

_behavsci, 2022, doi:10.3390/bs12100397_

Round 1

Reviewer 1 Report

Dear authors, I reviewed your article and I have some suggestions:

1. Read again the Instructions for authors and prepare the article according to it.

2. Lines 87-90 must be excluded.

3. At first use, write complete words and then use the abbreviation.

4. In table 1 check again the numbering.

5. Add more data and information at Results. It is too short with a lack of information.

6. Add Conclusion at the end of the article. There you can include last paragraph (lines 189-192) and add another information concerning your results.

Author Response

  1. Re-read the Instructions for Authors and prepare the article accordingly.

The manuscript has been reviewed according to the journal's instructions

  1. Lines 87-90 should be excluded.

The indicated lines have been excluded

  1. On the first use, write full words, and then use the abbreviation.

It has been corrected throughout the manuscript

  1. In Table 1 check the numbering again.

Verified

  1. Add more data and information in Results. It is too short with lack of information.

Results have been extended (lines 326-349)

  1. Add Conclusion at the end of the article. There you can include the last paragraph (lines 189-192) and add other information about your results.

The conclusions section has been added in which the main conclusions of the study are indicated and the previous lines 189-192 are added.

Reviewer 2 Report

The authors need to explain sample size? Based on 

what is 165 students included? Please calculate sample size.

The references must be arranged.

When the abbreviations are mentioned in the text for the first time the full text should be listed.

The title of the table and figure must be self-explanatory.

Author Response

  1. Do the authors need to explain the sample size? How much is 165 students included?

The participants section has been modified and this information has been detailed (lines 116-119).

  1. References should be sorted.

Each of the references has been revised, sorted and relisted

  1. When abbreviations are mentioned in the text for the first time, the full text should be included.

Corrected throughout the document

  1. The title of the table and figure should be self-explanatory.

Both titles have been corrected

Round 2

Reviewer 2 Report

Now, after fhe authors accepted suggestion, the manuscript is improved. After English language and style correction, it would be acceptible for publication.

Author Response

Thank you very much for your contributions, expression and style errors on the language have been corrected throughout the document to improve its quality by a specialist in translation and interpretation in English language that guarantees a level of C1 in English language proficiency. Changes are marked in green